# Nitrogen Removal from Landfill Leachate Using Biochar Derived from Wheat Straw

**DOI:** 10.3390/ma17040928

**Published:** 2024-02-17

**Authors:** Chinenye Adaobi Igwegbe, Michał Kozłowski, Jagoda Wąsowicz, Edyta Pęczek, Andrzej Białowiec

**Affiliations:** 1Department of Applied Bioeconomy, Wrocław University of Environmental and Life Sciences, Chełmońskiego 37A Str., 51-630 Wroclaw, Poland; ca.igwegbe@unizik.edu.ng (C.A.I.); jagusia.blueberry@gmail.com (J.W.); edyta.peczek@selena.com (E.P.); andrzej.bialowiec@upwr.edu.pl (A.B.); 2Department of Chemical Engineering, Nnamdi Azikiwe University, Awka 420218, Nigeria; 3Selena Industrial Technologies sp. z o.o., Pieszycka 3 Str., 58-200 Dzierżoniów, Poland

**Keywords:** adsorption, isotherm modeling, mechanism, wastewater treatment

## Abstract

Landfill leachate (LLCH) disposal poses challenges due to high pollutant concentrations. This study investigates the use of biochar (BC) derived from wheat straw for nitrogen content reduction. Laboratory experiments evaluated BC’s adsorption capacity (q_m_) for nitrogen removal from ammonium chloride solution (NH_4_Cl) and LLCH, along with testing isotherm models. The results demonstrated that BC was more efficient (95.08%) than commercial activated carbon AC (93.11%), the blank, in adsorbing nitrogen from NH_4_Cl. This superior performance of BC may be attributed to its higher carbon content (57.74%) observed through elemental analysis. Lower results for BC/LLCH may be due to LLCH’s complex chemical matrix. The Langmuir isotherm model best described BC/NH_4_Cl adsorption (q_m_ = 0.5738 mg/g). The AC/NH_4_Cl data also fitted into the Langmuir (R^2^ ˃ 0.9) with a q_m_ of 0.9469 mg/g, and 26.667 mg/g (R^2^ ˂ 0.9) was obtained for BC/LLCH; the BC/LLCH also gave higher q_m_ (R^2^ ˃ 0.9) using the Jovanovich model (which also follows Langmuir’s assumptions). The mean energy of the adsorption values estimated for the AC/NH_4_Cl, BC/NH_4_Cl, and BC/LLCH processes were 353.55, 353.55, and 223.61 kJ/mol, respectively, suggesting that they are all chemisorption processes and ion exchange influenced their adsorption processes. The Freundlich constant (1/n) value suggests average adsorption for BC/LLCH. The BC/LLCH data followed the Harkins–Jura model (R^2^: 0.9992), suggesting multilayered adsorption (or mesopore filling). In conclusion, biochar derived from wheat straw shows promising potential for landfill leachate remediation, offering efficient nitrogen removal capabilities and demonstrating compatibility with various adsorption models. This research also lays the groundwork for further exploration of other biochar-based materials in addressing environmental challenges associated with landfill leachate contamination.

## 1. Introduction

The advancement of our civilization presents numerous opportunities and has the potential to enhance our quality of life [1,2]. Unfortunately, that development also faces new challenges, for example, the richest and more developed societies usually generate more waste than the poorest. Furthermore, it should be expected that as our civilization continues to evolve, the global volume of waste will surge significantly. Consequently, waste management specialists are faced with the formidable task of restructuring waste management systems to cope with this escalating volume and complexity.

One approach to waste management involves the use of landfills for storing debris. Despite occasional negative perceptions, landfills remain the most straightforward and cost-effective method of municipal waste disposal when compared to alternative methods [3]. Landfills offer the potential to mitigate the release of environmentally hazardous compounds and enable controlled conditions for the decomposition process. However, improper landfill management can lead to the unintended release of toxic compounds into the environment [4,5], with the generation of potentially harmful leachate being a significant concern. Notably, in the European Union (EU), there has been a decrease in the volume of waste landfilled from 2015 to 2021 due to increased efforts in recycling and incineration [6]. Leachate typically comprises a mixture of substances, including environmentally harmful heavy metals, ammonia nitrogen, humic substances, and refractory organic matter [7,8,9,10]. It poses a threat as it can infiltrate soil, impact plant life, and contaminate groundwater [8,11,12], and if not monitored, it can have adverse effects on human and animal health [13,14], potentially containing pathogenic microorganisms, antibiotics, and microplastics [15]. Elevated nitrogen concentrations in water can result in eutrophication, oxygen depletion, adverse effects on aquatic ecosystems, and potential health risks, such as methemoglobinemia in infants and a suggested association with certain cancers [16,17,18].

Given the hazardous nature of compounds found in leachates, it is imperative to purify this substance prior to its release into the environment. Various methods are utilized for its purification, often combining biological, chemical, and physical approaches. Notable techniques include membrane filtration [19], which can be complemented by leachate evaporation using biogas production and combustion for energy generation [20], along with coagulation using the Fenton process [21]. Biological treatments are also gaining attention, with microorganisms playing a crucial role in waste decomposition within landfills [22]. Additionally, plants offer a promising avenue for removing pollutants from leachates, with floating wetlands being one such example utilized for phytoremediation purposes [23].

Another very interesting method of landfill treatment is the use of adsorption. However, the effectiveness of this method is highly dependent on the type of adsorbent and composition of leachate. For instance, activated carbon (AC) and zeolite are commonly used because of their unique surface properties [24]. Another noteworthy and essential material is biochar. Biochar (BC) is a biomaterial derived from various biomass sources [25,26], including forest residues, agricultural byproducts like tomato marc [27], waste tea leaves [28], and even algal waste [29]. Biochar production is more economical [30,31] and has less environmental impact than the production of activated carbon. Furthermore, well-chosen biochar production features can provide biochar with similar effects as activated carbon with a lower economic and environmental price of production [32]. Importantly, biochar finds diverse applications beyond leachate treatment; it is also utilized in agriculture, energy production, and even pharmaceuticals [33].

Biochar has emerged as a promising solution for treating landfill leachate [34,35], with the added benefit of its potential to be produced from biowaste materials, aligning seamlessly with the principles of a circular economy [36]. Our study involved conducting a comprehensive feasibility assessment to determine the practicality of utilizing biochar derived specifically from wheat straw biowaste for landfill leachate treatment. Known for its adsorption properties, biochar presents a viable alternative to traditional activated carbon. While several investigations have explored the efficacy of biochar for nitrogen removal, particularly derived from wheat straw, our study specifically targets a critical gap in the distinct adsorption capacity of wheat straw-derived biochar for nitrogen removal from landfill leachate. The results of this study will provide valuable information on the viability and limitations of using biochar from biowaste for landfill leachate treatment.

## 2. Materials and Methods

### 2.1. Material Collection and Preparation

Wheat straw (Figure 1a) was collected from the Wrocław University of Environmental and Life Sciences Research and Didactic Field Station in Swojczyce, Wrocław, Poland. The wheat straw was used as a feedstock to produce biochar (BC) using the pyrolysis process. The straw, which contained approximately 8% moisture, was dried and ground in a mill with a sieve size of 1 mm. The resulting material was then subjected to pyrolysis at 600 °C for an hour in a muffle furnace (8.1/1100; SNOL, Utena, Lithuania) in an oxygen-limited atmosphere, and the cooled biochar (Figure 1b) was a black, light powder. The BC powder was stored in airtight containers to avoid any reaction before the adsorption and characterization experiments. The initial mass of the dry straw after milling was 224.8 g, and the final mass of the produced biochar was 150.7 g, representing a loss of 33% of mass during pyrolysis.

To compare the adsorption properties of BC and activated carbon (AC), AC granules (Figure 2) with a diameter of 4 mm were ordered from “trzmiel.com.pl”. The manufacturer reported that the product was in the form of granules with varying lengths and was commonly used in ponds and marine and freshwater aquariums to neutralize odors, chemical impurities, and discoloration without affecting water parameters. BC and AC were separately ground in a mortar into smaller particles for the adsorption experiments. 

All reagents including ammonium chloride (NH_4_Cl) and distilled water were purchased from MERCK Sp. z o. o., Warsaw, Poland. All the chemicals used were of analytical grade and were used without further purification. Prior to conducting experiments, all laboratory equipment was calibrated according to standard protocols. The chemicals were collected and prepared per established protocols. The experiments were carried out in triplicate to ensure the reliability of the results.

### 2.2. Adsorbent (BC and AC) Property Determination

The properties of the adsorbent materials were examined to observe their adsorptive characteristics. The functional groups present in AC and BC were identified using the KBr pressed disk technique, via Fourier-transform infrared spectroscopy (FTIR, PerkinElmer Spectrum IR 10.7.2, Waltham, MA, USA). Additionally, the Brunauer–Emmett–Teller (BET) surface area and pore volume of the BC was determined by measuring the nitrogen adsorption isotherm using a gas absorption analyzer (Quantachrome—Autosorb iQ-MP-MP (Viton), Anton Paar Analyzer, Boynton Beach, FL, USA) at 77K. Finally, elemental and ultimate analyses were conducted to gain further insights into the materials’ composition and properties using an elemental analyzer (2400 Series II; PerkinElmer, Waltham, MA, USA).

### 2.3. Landfill Leachate Collection

The landfill leachate (LLCH) (Table 1) used in this study was collected from the municipal waste landfill in Zgorzelec, Poland (51.193829 N, 15.0161520 E), which has been operating since 2007 and is managed by the Municipal Enterprise of Waste Management. The landfill is situated approximately 500 m from residential buildings and 200 m from a nearby river. The landfill site spans across quarters covering a total area of 1.35 hectares and is designed to accommodate up to 30,000 metric tons of waste annually. The secured quarters have a capacity of 128,000 Mg and are equipped with a drainage system to discharge landfill effluents at its lower points. The low biodegradability index of leachate (BOD_5_/COD = 0.23) indicates that it contains a significant proportion of refractory organic matter which may be difficult to treat with biological treatment methods [37]. In order to address this issue, operators of wastewater treatment plants can monitor this biodegradability index to optimize the treatment process and achieve better treatment efficiency. This may involve implementing physicochemical processes such as adsorption techniques for the treatment of this type of water.

### 2.4. Adsorption Experiments

The possible replacement of AC using BC was investigated for nitrogen adsorption capacity. The research consisted of two stages. In the first stage, all adsorbents (activated carbon [AC] and biochar [BC]) were tested using an aqueous solution of ammonium chloride (NH_4_Cl) at variable concentrations (100, 80, 60, 40, and 20 mg/L), and their performance was compared. The NH_4_Cl solution was prepared by dissolving a known mass of NH_4_Cl in distilled water. In the second stage, the produced BC was tested for nitrogen adsorption capacity using landfill leachate (LLCH) solutions. The LLCH adsorption tests were conducted using BC only, and the chosen initial concentrations of the leachate solution were 87.08, 174.16, 261.24, 348.32, and 435.4 mg/L. 

Initially, the experiment was conducted for both adsorbents (AC and BC) on NH_4_Cl solutions at concentrations of 100, 80, 60, 40, and 20 mg/L. For each concentration of NH_4_Cl, tests were performed for each adsorbent to obtain average data. Erlenmeyer flasks containing 2 g of the given adsorbent and 50 mL of the NH_4_Cl solution were placed on a shaker (Figure 3a) for 24 h at 90 revolutions per minute (rpm) and room temperature (25 ± 0.5 °C). After shaking, the contents of the flasks were filtered (Figure 3b) and tested for nitrogen content.

In the final test, biochar from the straw (BC) was shaken with LLCH, and tests were conducted for each concentration of the leachate solution. The concentrations tested were 87.08, 174.16, 261.24, 348.32, and 435.4 mg/L, which corresponded to 20%, 40%, 60%, 80%, and 100% LLCH dilutions, respectively, with distilled water. Each Erlenmeyer flask contained 2 g of BC and 50 mL of the LLCH concentration solution. The shaker was set to 90 rpm, and the duration was 24 h. Samples of biochar and filtrate were filtered and tested. Nitrogen concentration was determined in all samples using an Enviro TOC cube liquid sample analyzer (Elementar Analysensysteme GmbH, Langenselbold, Germany) to investigate the potential of BC treatment for LLCH disposal.

The average final concentrations of nitrogen from the filtrate ($C_{e}$) were determined and the average sorption capacity in mg N/g of adsorbent ($q_{e}$) was determined as follows (Equation (1)).
(1)$$q_{e=\frac{\left(C_{i}-C_{e}\right)V}{M}}$$
where $C_{i}$= the initial nitrogen concentration (mg/L) in aqueous solution of NH_4_Cl and LLCH, V = the volume of the adsorbate solution (L) treated, and M = the mass of the adsorbent (BC or AC) (g). 

### 2.5. Adsorption Isotherm Model Study

Adsorption isotherms are an essential tool in wastewater treatment as they provide information about the adsorption capacity, efficiency, and optimization of adsorption. They also provide valuable information about the strength and nature of the interaction between the adsorbent and the contaminant in the wastewater [49,50]. Moreover, adsorption isotherm data can be used to compare the adsorption capacity of different adsorbents used in the wastewater treatment process [51,52]. This comparison can help in choosing the most effective adsorbent for a specific application; it can also be used to optimize the adsorption process and improve its efficiency. The data from the AC/NH_4_Cl, BC/NH_4_Cl, and BC/LLCH adsorption processes were modeled based on the following isotherms: Langmuir (Equation (2)) [53], Temkin (Equation (3)) [54], Freundlich (Equation (4)) [55], Harkins–Jura (Equation (5)) [56], Jovanovich (Equation (6)) [57], and Dubinin–Radushkevich (Equations (7)–(9)) [58].
(2)$$\mathrm{Langmuir}\;\mathrm{model}:\;\frac{C_{e}}{q_{e}}=\frac{1}{q_{m}K_{L}}+\frac{C_{e}}{q_{m}}$$
(3)$$\mathrm{Temkin}\;\mathrm{model}:\;q_{e}=B_{T}\;Ln\;A_{t}+B_{T}\;Ln\;C_{e}$$

(4)Freundlich model: Log ⁡qe=1nLog⁡Ce +Log⁡KF (5)$$\mathrm{Harkins}–\mathrm{Jura}\;\mathrm{model}:\;\frac{1}{q_{e}^{2}}=\left(\frac{B}{A}\right)\;-\;\left(\frac{1}{A}\right)logC_{e}$$(6)$$\mathrm{Jovanovich}\;\mathrm{model}:\;In\;q_{e}=In\;q_{max}-K_{J}C_{e}$$(7)$$\mathrm{Dubinin}–\mathrm{Radushkevich}\;\mathrm{model}:\;Lnq_{e}=Lnq_{D}-B\varepsilon^{2}$$(8)$$\varepsilon =RTLn\left(1+\frac{1}{C_{e}}\right)$$(9)$$E=\frac{1}{{\left(2B\right)}^{1/2}}$$
where *q_e_* = the amount of adsorbate adsorbed on the adsorbent at equilibrium (mg/g), *q_m_* = the maximum monolayer adsorption capacity (mg/g), and *K_L_* = the Langmuir constant which is related to the heat of adsorption (L/mg); *K_F_* = the Freundlich isotherm constant (L/g), *n* = the magnitude of favorability of adsorption, and *q_e_* = the amount of adsorbate adsorbed at equilibrium (mg/g); *A_T_* = the Temkin constant and *B_T_* = the constant associated with the heat of adsorption; *K_J_* = the Jovanovich constant; *B* and *A* = Harkins–Jura constants; and *q_D_* = the theoretical saturation capacity (mg/g), *B* = the constant related to the mean free energy of adsorption per mole of the adsorbate (mol^2^/J^2^), *ε* = the Polanyi potential which is associated with equilibrium, *R* = the universal gas constant (8.314 J/mol/K), *T* = the temperature in Kelvin, and *E* = the mean sorption energy (kJ/mol).

Using the Dubinin–Radushkevich isotherm, the apparent adsorption energy may be determined. Based on this sorption energy, *E* (Equation (9)), one may determine if an adsorption process involves chemisorption or physisorption. If the activation energy is less than 8 kJ/mol or between 8 and 16 kJ/mol, the adsorption is either physisorption or chemisorption [59,60]. The value of the correlation coefficient (R^2^) was used to determine which model better characterized the data.

## 3. Results and Discussion

### 3.1. Properties of the Commercial Activated Carbon and Straw Biochar

Table 2 shows the results of the elemental analysis performed on AC and BC. It provides insights into the materials’ composition and properties of the adsorbents used for the adsorption studies.

The final elemental composition of the AC sample was the following: carbon (C) content of 37.51%, hydrogen (H) content of 1.19%, nitrogen (N) content of 2.52%, sulfur (S) content of 0.42%, and oxygen (O) content of 58.36%. The elemental composition of the BC sample, on the other hand, was the following: C content of 57.74%, H content of 1.85%, N content of 0.51%, S content of 0.46%, and O content of 39.44%. These findings show a considerable difference in the carbon content between AC and BC, with BC having a much greater carbon content. This disparity shows that BC will have a higher adsorption capability for contaminants than commercial AC. Furthermore, BC had lower nitrogen, sulfur, and oxygen contents than AC. These differences in elemental composition are believed to contribute to the two materials’ different adsorption capacities. Carbon increases adsorption capacity and affinity for various compounds, making carbon-rich adsorbents widely applicable for various adsorption applications such as water and air purification and gas separation [61].

FTIR analysis provides valuable information regarding the chemical and functional groups present in materials. These could offer insights into the potential interactions and adsorption mechanisms between the AC and BC materials and the adsorbate, nitrogen. The transmittance (T) versus wavelength plots for the functional groups present in the AC and BC are depicted in Figure 4a,b, respectively.

In Figure 4a, a prominent peak observed at 743.97 cm^−1^ indicates the presence of C=C bending in alkene bonds. Additionally, a peak at 1017.74 cm^−1^ corresponds to the C–O stretching characteristic of primary alcohols within the AC sample. The presence of alkene bending, indicating unsaturated carbon–carbon double bonds (π bonds) [62], can contribute to the overall surface properties of an adsorbent material and potentially affect its affinity for nitrogen adsorption. The C–O stretching in primary alcohols signifies the presence of hydroxyl (OH) functional groups. Hydrogen bonding interactions can enhance adsorption by increasing the affinity and retention of adsorbates on a material’s surface [63]. Nitrogen molecules can form hydrogen bonds with these hydroxyl groups, potentially enhancing the adsorption of nitrogen on the adsorbent surface [64]. 

Several significant peaks can be observed in Figure 4b, each corresponding to distinct functional groups within the BC. The peak at 786.83 cm^−1^ signifies the C–Cl stretching in alkyl halides. At 1021.16 cm^−1^, a peak can be observed, indicating the C–N stretching in aliphatic amines. Furthermore, the peak at 2694.47 cm^−1^ represents the H–C=O:C–H stretching characteristic of aldehydes present in BC. The C–Cl stretching in alkyl halides indicates the presence of halogenated organic compounds which can enhance the interaction between nitrogen molecules and the adsorbent through dipole–dipole or charge-induced interactions [65]. C–N stretching occurs in aliphatic amines (amino functional groups). Nitrogen adsorption may be influenced by the presence of amine groups through chemical complexation or ion exchange mechanisms, thereby forming coordination bonds with nitrogen molecules [66,67], facilitating their adsorption. The H–C=O:C–H stretching in aldehydes indicates the presence of carbonyl (C=O) groups. Nitrogen adsorption may be influenced by these carbonyl groups through weak interactions, such as van der Waals forces or dipole-induced dipole interactions [68].

Figure 5 illustrates the outcomes of the BET analysis performed on the BC sample. The blue curve represents the adsorption isotherm (quantity of gas adsorbed at different pressures) and the red BET curve plot is the linear region derived from the adsorption isotherm used to determine monolayer capacity and specific surface area. The blue curves represent the experimental values. The observed adsorption isotherm aligns with the characteristics of Type IV isotherms, which typically indicate multilayered adsorption followed by capillary condensation on mesoporous materials. The biochar exhibited a specific surface area (S_BET_) with a value of 12 m^2^/g, indicating its capacity for surface interactions and adsorption processes. Moreover, the pore volume (V_T_) was measured at 0.0197 cm^3^/g, indicating the presence of mesopores and suggesting potential storage capacity for various molecules or ions [69]. This characterization also revealed a mean pore size (L) of 3.28 nm, which is between 2 and 50 nm [70], indicating the predominance of mesopores in the biochar structure. These structural properties highlight the biochar’s potential for adsorption-based applications, which can have significant implications for environmental remediation.

### 3.2. Adsorption Tests with Ammonium Chloride Solution for the AC and BC

Table 3 displays the results of nitrogen adsorption employing the commercial AC and BC from the NH_4_Cl solution. The plots in Figure 6 illustrate the correlations between the qe and initial solution concentrations. 

After executing the first step of the research using the NH_4_Cl solution, the results indicated that the adsorbents (AC and BC) utilized were comparable (Table 3). Both the qe and nitrogen removal (R) increased with increasing nitrogen content when AC and BC were used. The maximum qe of 2.33 mg N/g and R of 93.11% were determined for AC; meanwhile, the maximum qe of 2.38 mg N/g and R maximum of 95.08% were determined for BC [71]. The increase in adsorption capacity with increasing nitrogen concentration may be due to a faster adsorption rate or uptake and use of all active sites accessible for adsorption at higher concentrations or to a larger availability of nitrogen ions in the solution for adsorption [72,73].

The results suggest that the active sites on the AC and BC were not entirely depleted at lesser concentrations, which is consistent with the findings of previous studies [72,73]. These results indicate that both AC and BC have the potential for the removal of nitrogen from wastewater, but the use of BC gave a higher *qm* and efficiency; this may be due to the fact that BC contained a higher carbon content than AC, which was observed through the elemental analysis in Table 2. However, further investigation was conducted to determine the feasibility of using BC adsorbent for the removal of nitrogen from real wastewater samples (leachate) in this study.

To support these findings, previous studies have reported that biochar derived from various feedstocks, including straw, can effectively adsorb nitrogen from aqueous solutions [74]. It should be noted that the results obtained in this study are consistent with those reported by other researchers. For example, Werkneh, Habtu, and Beyene [72] found that the adsorption capacity of AC increased with increasing nitrogen concentration. Similarly, Sampranpiboon, Charnkeitkong, and Feng [73] reported that the adsorption of nitrogen from wastewater using BC increased with increasing nitrogen concentration. 

### 3.3. Adsorption Test Results for Biochar on Landfill Leachate

Table 4 shows the results of nitrogen adsorption using biochar derived from wheat straw for LLCH treatment. The correlation between the calculated qe and initial LLCH concentrations is illustrated in Figure 7.

The results of the second phase of the LLCH investigation differed significantly from those obtained using ammonium chloride in Section 3.2. The maximum adsorption capacity for nitrogen was 10.09 mg/g of BC, and the highest Ce concentration was obtained with a starting nitrogen concentration of 435.4 mg/L. The increase in qe at higher leachate concentrations suggests that a stronger force drives the mass from the liquid solution to the biochar, similar to the AC/NH_4_Cl and BC/NH_4_Cl sorption processes [73,75,76]. The lowest Ce (lower concentration of nitrogen remaining after adsorption) was obtained, with a maximum efficiency (Rmax) of 92.74%, at the lowest concentration (178.3 mg/L, 20% LLCH). The BC effectively improved the liquid’s color and eliminated contaminants such as nitrogen. A comparison of the colors of LLCHs before and after the adsorption procedure is shown in Figure 8a–e. These suggest that the greater the dilution (or concentration decrease) of the LLCH, the better the elimination of color and nitrogen content [72].

Furthermore, the effectiveness of biochar in removing contaminants such as nitrogen from landfill leachate has also been demonstrated [77]. Additionally, studies have shown that the adsorption capacity of biochar increases from increasing the initial concentration of nitrogen or other contaminants in the solution [78]. Tests of biochar on LLCH yielded lower results due to the much more complex chemical matrix of the LLCH.

### 3.4. Results for Adsorption Isotherms

The results obtained from the adsorption isotherm modeling are summarized in Table 5. Detailed modeling plots can be found in the Appendix A. The determination coefficients (R^2^) were used to assess the goodness of fit for each isotherm model.

The Freundlich isotherm provided the best fit for the AC/NH_4_Cl adsorption in terms of its R^2^ value (Table 5). This indicates that the adsorption was of a multilayered type and that the AC’s surface is highly heterogeneous [79]. This is typical of adsorbents manufactured using thermochemical methods [80,81]. Additionally, the adsorption mechanism reveals that adsorbents are adsorbed into stronger binding sites first and then to binding sites with lower energy [82]. When site occupation increases, the strength of binding diminishes. In every instance, the data adhered to the Freundlich isotherm. This compatibility of the data with the Freundlich model denotes a chemisorption process. Freundlich’s parameter, n, determines the intensity of adsorption; if the n values are within the range of 0.1 < 1/*n* < 0.5, it indicates good and promising adsorption, whereas 0.5 < 1/*n* < 1 and 1/*n* > 1 indicate moderate and weak adsorption, respectively [83]. The 1/*n*-values for BC/NH_4_Cl and AC/NH_4_Cl (Table 5) show weak adsorption, but the 1/*n*-value for BC/LLCH suggests average adsorption [84].

The Langmuir isotherm was the best fit for the BC/NH_4_Cl adsorption. Thus, it can be said that there is a monolayered coverage of nitrogen over the homogeneous surface of BC [85,86] with a monolayered adsorption capacity and q_m_ of 0.5738 mg/g, which indicates homogenous adsorption. Because each site can only retain one adsorbate, there is no interaction between the adsorbed molecules [86]. Though the Langmuir isotherm was not the best fit, it is also accurate for modeling adsorption data. The AC/NH_4_Cl data also fit into the Langmuir (R^2^ ˃ 0.9) with a q_m_ of 0.9469 mg/g, and 26.667 mg/g was obtained for BC/LLCH. The adsorptive capacity determined by the Langmuir model for the real leachate was higher than that of the synthetic leachate using BC (Table 5). This shows the possible replacement of AC with biochar from biowaste. The Harkins–Jura isotherm was the best fitted model for the BC/LLCH adsorption data with an R^2^ of 0.9992; this proves its validity for solute adsorption and suggests multilayered adsorption (mesopore filling [69], also confirmed from the BET analysis). In the same way, Tunç and Duman [87]‘s experimental data fit the Harkins–Jura model better (R^2^ ˃ 0.99) than any of the other 14 isotherm models used in their study. Monolayer adsorption capacity can also be determined using the Jovanovich model, which also gave a high R^2^ ˃ 0.9, but the Langmuir model gave higher monolayer adsorption capacities. The Jovanovich model also follows Langmuir’s assumptions of monolayered adsorption on homogeneous surfaces [88]. Also, the mean energy of sorption values *E* were estimated as 353.55, 353.55, and 223.61 kJ/mol for AC/NH_4_Cl, BC/NH_4_Cl, and BC/LLCH, respectively, suggesting that they are chemisorption processes since their values were found to be less than 8 kJ/mol. Also, this implies that intraparticle diffusion predominated and that ion exchange influenced the adsorption processes [89,90,91].

### 3.5. Interpretation and Practical Considerations

Our investigation into nitrogen adsorption using biochar derived from wheat straw has unveiled promising results with implications for both environmental remediation and sustainable waste management. However, it is essential to acknowledge the inherent technological constraints and limitations that might influence the practical applicability of our findings. The complex chemical matrix of landfill leachate presents challenges in achieving consistent results, as highlighted by variations observed in the second phase of the investigation compared to the ammonium chloride experiments (Section 3.3). These nuances warrant careful consideration when translating laboratory-scale successes into real-world scenarios. Environmental concerns also play a crucial role in the adoption of biochar-based solutions. While our study demonstrates the efficacy of biochar in removing contaminants, we recognize the need for a deeper understanding of potential secondary effects, especially when considering large-scale implementation. Addressing these concerns requires a holistic approach, considering the broader environmental impact and sustainability of biochar applications. Technological challenges, such as the variability in adsorption capacities under different conditions, require attention. The isotherm modeling in this study (Section 3.4) provides valuable insights into adsorption mechanisms, but translating these models into practical applications requires careful consideration of operational parameters. Considering these challenges will provide a foundation for future research directions in optimizing biochar-based adsorption processes.

### 3.6. Molecular Interactions and Adsorption Mechanisms

To gain a comprehensive understanding of the adsorption mechanisms observed in our study, we present a detailed analysis of the molecular interactions at the adsorbate–adsorbent interface.

#### 3.6.1. Adsorption on AC with Ammonium Chloride

Upon dissociation, ammonium chloride yields ammonium ions (NH₄⁺) and chloride ions (Cl^−^). These ions engage with the charged surface of activated carbon through ion–dipole interactions [92], where functional groups on the activated carbon surface attract and interact with the ions. The charged nature of AC and NH₄⁺/Cl^−^ ions induces attractive electrostatic forces, facilitating adsorption through positive and negative charges on the surface and ions, respectively. Chemical bonds may form between distinct functional groups on activated carbon and NH₄⁺/Cl^−^ ions (chemisorption, which was also observed from the isotherm study), involving covalent bonds or strong interactions. Oxygen-containing functional groups on the AC surface facilitate such chemical bonding.

Van der Waals forces, specifically London dispersion forces, contributed to the adsorption process by attracting molecules to the activated carbon surface. These non-polar interactions arise from temporary fluctuations in electron distribution and are influenced by induced dipole moments. Functional groups such as OH, N–H, or C=O on the AC surface can participate in dipole–dipole interactions [92] since ammonium chloride is an ionic compound. The porous structure of activated carbon, comprising micropores and mesopores, enables physical interactions and potential pore filling [63]. Micropores selectively adsorb certain ions, while larger mesopores accommodate ion clusters.

In addition, the adsorption of nitrogen from ammonium chloride onto AC is further influenced by hydrophobic interactions including London dispersion (ACs are hydrophobic in nature and have a C=C double bond and C–O [93]). The unsaturated carbon–carbon double bonds (π bonds) contribute to π–π electron interactions [94]. Oxygen atoms in C–O groups can also participate in hydrogen bonding with polar groups [92] in ammonium chloride. The oxygen atom in the OH group can act as a hydrogen bond acceptor [95]. Figure 9 provides a visual summary of these complex molecular interactions.

#### 3.6.2. Adsorption on Biochar (BC) with Landfill Leachate (LLCH)


Considering the isotherm models collectively studied in Section 3.4, the data support the conclusion that the adsorption of nitrogen from landfill leachate onto activated carbon involves both physisorption and chemisorption processes. The Langmuir model suggests monolayered coverage, while the Freundlich and Harkins–Jura models indicate a heterogeneous surface and multilayered adsorption. The Jovanovich model further supports the idea of monolayered adsorption on a homogeneous surface.

Moreover, the following functional groups were observed in the BC: C–Cl, C–N, H–C=O, C–H, and C=O. The nitrogen compounds observed in the leachate were nitrates (NO₃^−^) and nitrites (NO₂^−^) (Table 1). The electronegativity of chlorine in the C–Cl bond within biochar may lead to van der Waals [96] or dipole–dipole interactions with nitrogen species [92] from LLCH. The C–N bond in biochar may engage in π–π or dipole–dipole interactions with nitrogen-containing compounds in the leachate, such as amines or amides [92]. Hydrogen bonding interactions involving the hydrogen in the H–C=O group may occur between biochar and polar nitrogen-containing groups [97]. Meanwhile, weaker C–H bonds in biochar can contribute to hydrophobic interactions with non-polar nitrogen compounds in the leachate [98]. Biochar’s carbonyl groups (C=O) may form hydrogen bonds with polar nitrogen compounds and engage in π–π interactions with aromatic nitrogen-containing compounds [99,100]. The porous structure of the BC observed from the BET analysis enables physical interactions and potential pore filling [63]. Similar to the AC/NH_4_Cl system, the adsorption of nitrogen from landfill leachate onto BC is driven by a combination of electrostatic attraction, hydrogen bonding, hydrophobic interactions, π–π electron donor interaction, and Van der Waals forces (Figure 9). The complex chemical matrix of LLCH introduces additional complexities to the adsorption process. Figure 9 visually illustrates the intermolecular interactions contributing to the efficient adsorption of nitrogen onto BC from landfill leachate.

### 3.7. Comparative Assessment with Previous Biochar Studies on Nitrogen Removal

Several investigations have delved into the efficacy of biochar, particularly derived from wheat straw, as a promising adsorbent for nitrogen removal. Our research, however, stands out by focusing exclusively on biochar sourced from landfill leachate straw (LLCH). While certain characteristics like porosity and surface area may be shared across different biochars, the origin of the feedstock and resulting chemical composition significantly influence their adsorption behaviors. In contrast to previous studies, our work delineates the distinct adsorption capacity of LLCH-derived biochar for nitrogen removal from landfill leachate. Detailed findings, as outlined in Table 4 and Figure 7 and Figure 8, highlight a noteworthy maximum adsorption capacity of 10.09 mg/g at an initial nitrogen concentration of 435.4 mg/L, accompanied by a BET surface area of 12 m^2^/g.

Surface modification plays a crucial role in enhancing biochar adsorption capacity, echoing similar studies on wheat straw-derived biochar. For example, biochar produced from wheat straw, subjected to low-temperature pyrolysis at around 450 °C, underwent activation with HCl and was coated with varying iron amounts (FeCl₃·6H₂O) [101]. The results showcased a substantial increase in biochar adsorption capacity post-HCl activation and iron coating, resulting in an impressive specific surface area (SBET) of 138.56 m²/g. Optimal iron-to-biochar ratios were identified, underlining the potential for surface modification for tailoring adsorption characteristics. The maximum adsorption capacity was 2.47 mg/g.

In another notable study, Mg–Fe-layered double hydroxide (MgFe-LDH) particles were introduced into wheat straw biochar through liquid-phase deposition [102]. The resulting biochar/MgFe-LDH composite demonstrated a robust sorption ability for nitrate in aqueous solutions, boasting a Langmuir maximum adsorption capacity of 24.8 mg/g with a MgFe-LDH dosage of 2 g/L. The composite exhibited high selectivity for nitrate, even in the presence of sulfate and phosphate. In summary, our research offers unique insights into the nitrogen removal capabilities of LLCH-derived biochar, emphasizing the role of feedstock origin and surface modification in tailoring biochar for enhanced adsorption performance.

## 4. Conclusions

Leachates produced in landfills pose a threat to the environment. The use of the adsorption process is one of the methods of their disposal, and activated carbon (AC) is a known adsorbent used in the above-mentioned process. Their replacement can be biochar (BC) from biowaste. The biochar derived from wheat straw was first tested for nitrogen removal from ammonium chloride (NH_4_Cl) solution and compared with AC. Then, this study explored the potential of the wheat straw-derived biochar for nitrogen removal from landfill leachate (LLCH), to address a pressing environmental concern. Through a comprehensive experimental investigation, we elucidated the adsorption mechanisms and determined the optimal conditions for nitrogen removal using these adsorbents. 

The results demonstrated that both biochar and activated carbon exhibit high efficiency in adsorbing nitrogen from ammonium chloride solutions, with biochar showing a particularly promising performance. The biochar showed great efficiency on the LLCH. The Langmuir, Temkin, Freundlich, Harkins–Jura, Jovanovich, and Dubinin–Radushkevich isotherm models were employed to characterize the adsorption behaviors for the AC/NH_4_Cl, BC/NH_4_Cl, and BC/LLCH processes, revealing monolayered coverage and multilayered adsorption on the surfaces of the adsorbents. Furthermore, molecular interaction analyses elucidated the complex mechanisms driving the adsorption process, including ion–dipole interactions, hydrogen bonding, hydrophobic interactions, and Van der Waals forces. These insights contribute to a deeper understanding of the adsorption process and provide a foundation for future research in optimizing biochar-based adsorption systems.

The findings of this study have significant implications for environmental remediation and waste management in mitigating environmental pollution. By elucidating the adsorption capacities and mechanisms of biochar and activated carbon, this research offers a viable solution for mitigating nitrogen contamination in landfill leachate. Moreover, the use of biochar derived from agricultural waste presents an eco-friendly alternative to conventional adsorbents, reducing reliance on non-renewable resources and mitigating the environmental impact of waste disposal. The scalability and cost-effectiveness of biochar production will further enhance its potential for widespread adoption in landfill leachate treatment facilities.

## Figures and Tables

**Figure 1 materials-17-00928-f001:**
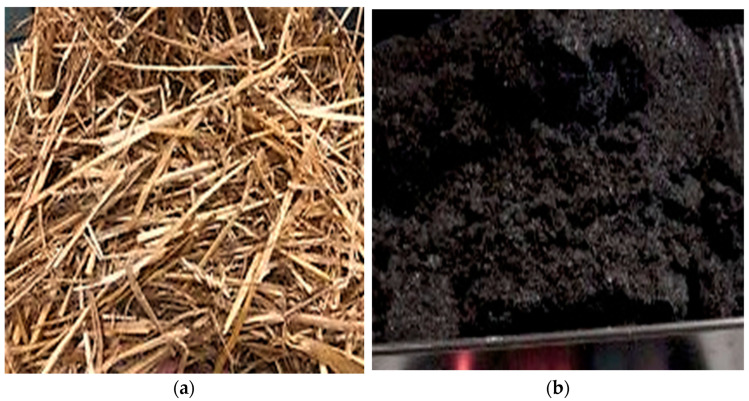
(**a**) Wheat straw used in this study; (**b**) Biochar from the wheat straw.

**Figure 2 materials-17-00928-f002:**
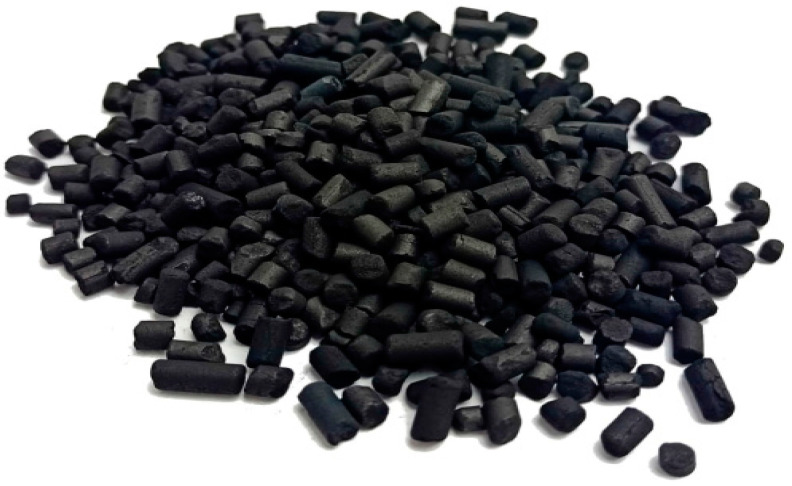
Activated carbon used in the experiment.

**Figure 3 materials-17-00928-f003:**
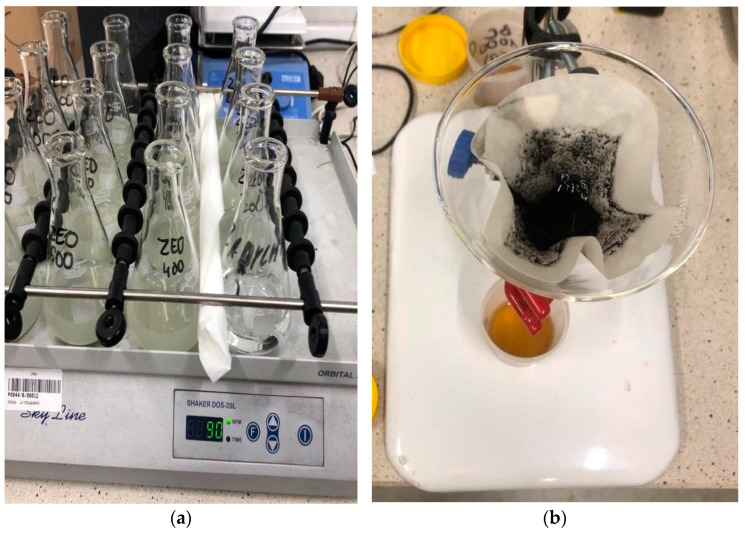
(**a**) Samples of solution and adsorbent on a shaker; (**b**) Filtration of the contents of the flasks after adsorption tests through filter paper.

**Figure 4 materials-17-00928-f004:**
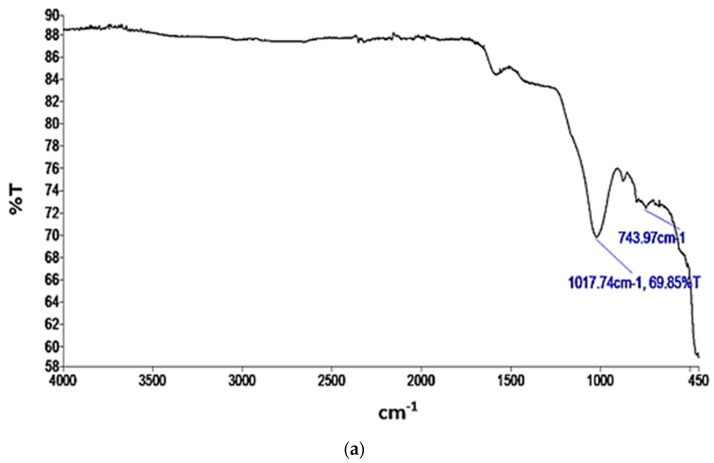
IR spectra of (**a**) AC; (**b**) BC.

**Figure 5 materials-17-00928-f005:**
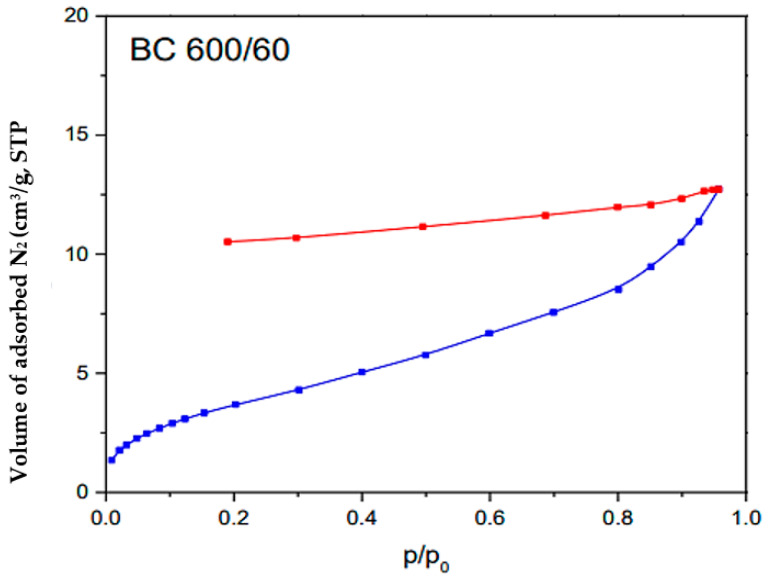
BET analysis of the BC.

**Figure 6 materials-17-00928-f006:**
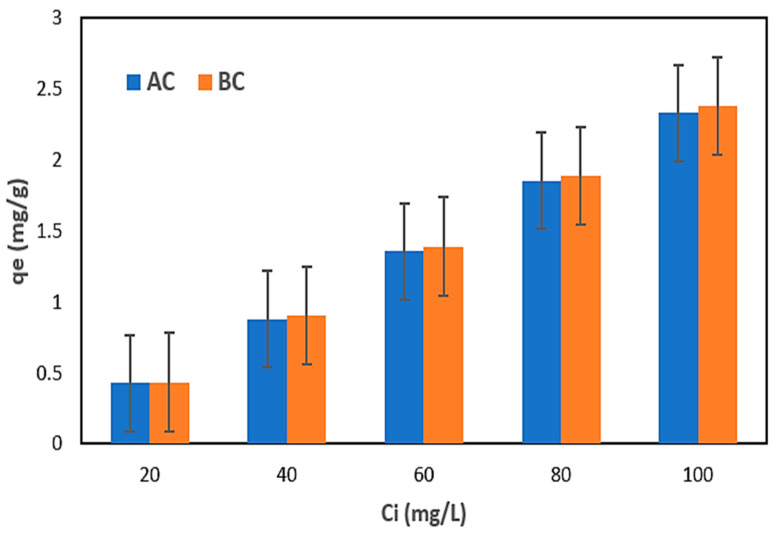
Dependence of the qe on the initial concentration of nitrogen in the NH_4_Cl solution using AC and BC.

**Figure 7 materials-17-00928-f007:**
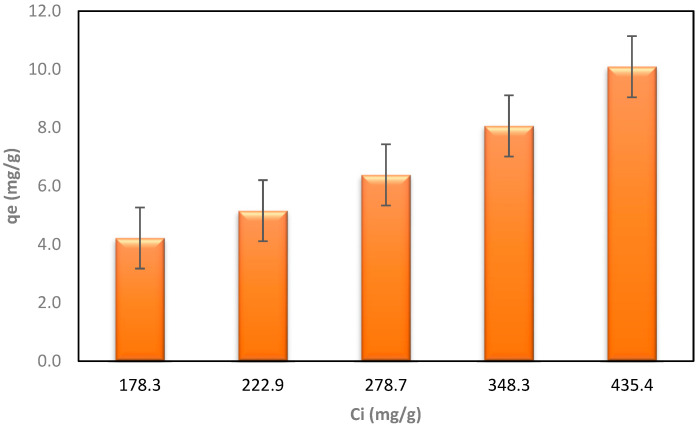
The dependence of the qe after the process on the initial concentration of LLCH.

**Figure 8 materials-17-00928-f008:**
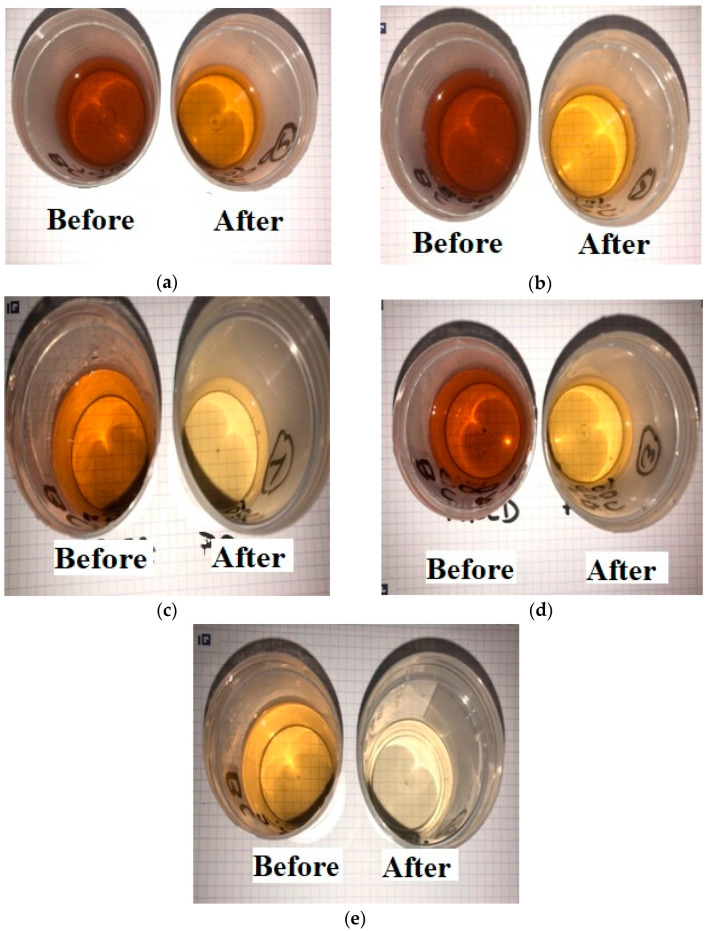
Colors of LLCH at (**a**) 100%; (**b**) 80%; (**c**) 60%; (**d**) 40%; (**e**) 20%. LLCH dilutions before and after the adsorption process.

**Figure 9 materials-17-00928-f009:**
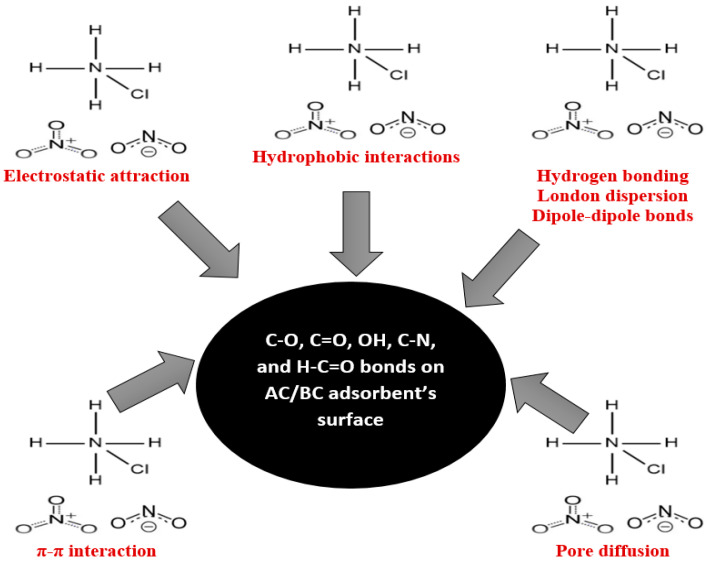
Summary of the adsorption mechanism of nitrogen onto AC from ammonium chloride and onto BC from landfill leachate.

**Table 1 materials-17-00928-t001:** Properties of the LLCH.

Parameter	Unit	Analysis Methodology	Result	Measurement Uncertainty
Biochemical oxygen demand (BOD_5_)	mg/L O_2_	PN-EN ISO 5815-1:2019-12 [38]	270	“±50”
Chemical oxygen demand (COD)	mg/L O_2_	PN-ISO 15705:2005 pkt 10.2 [39]	1150	“±160”
General suspensions	mg/L	PN-EN 872:2007/Ap1:2007 [40]	13	“±3”
Total nitrogen	mg/L	PN-EN 12260:2004 [41]	430	“±90”
Total phosphorus	mg/L	PN-EN ISO 11885:2009 [42]	2.8	“±0.4”
General solutes	mg/L	PN-78/C-04541 [43]	7130	“±713”
Ignition losses	mg/L	PN-78/C-04541 [43]	5520	“±552”
Ammonium ion/ammonia	mg/L	PN-EN ISO 11732:2007 [44]	>130	–
Nitrates	mg/L	PN-EN ISO 13395:2001 [45]	>445	–
Nitrite	mg/L	PN-EN ISO 13395:2001 [45]	3.1	“±0.5”
pH	–	PN-EN ISO 10523:2012 [46]	7.9	“±0.2”
Electrical conductivity (25 °C)	µS/cm	PN-EN 27888:1999 [47]	10,560	“±528”
Chloride	mg/L	PN-ISO 9297:1994 [48]	1620	“±240”

**Table 2 materials-17-00928-t002:** Elemental analysis of AC and BC.

Material	%C	%H	%N	%S	%O
AC	37.51	1.19	2.52	0.42	58.36
BC	57.74	1.85	0.51	0.46	39.44

**Table 3 materials-17-00928-t003:** The results of the nitrogen content analysis in filtrates of AC and BC and other parameters using NH_4_Cl solutions.

(Ci), mg/L	(Ce), mg/L	(qe), mg N/g	(R), %
AC sample			
20	2.94	0.43	85.28
40	4.80	0.88	88.01
60	6.41	1.34	90.26
80	5.87	1.85	92.67
100	6.89	2.33	93.11
BC sample			
20	2.60	0.44	87.02
40	4.21	0.89	90.45
60	4.40	1.39	92.67
80	3.89	1.90	94.46
100	4.92	2.38	95.08

**Table 4 materials-17-00928-t004:** The results of the nitrogen content analysis in the LLCH filtrates after BC usage.

(C_i_), mg/L	(Ce), mg/L	(qe), mg/g
178.3	9.64	1.94
222.9	16.71	3.94
278.7	23.38	5.95
348.3	25.83	8.06
435.4	31.63	10.09

**Table 5 materials-17-00928-t005:** Isotherm parameters for the adsorption processes.

Isotherm	Parameters	AC/NH_4_Cl	BC/NH_4_Cl	BC/LLCH
Langmuir	q_m_ (mg/g)	0.9469	0.5738	26.667
	K_L_ (L/mg)	0.1044	0.1647	0.0163
	R^2^	0.9179	0.9574	0.4013
Temkin	B_T_ (J/mg)	40.004	51.764	30.547
	A_T_	0.3781	0.4192	0.1180
	R^2^	0.7044	0.6793	0.8183
Freundlich	1/n	1.3335	2.6116	0.7069
	K_F_ (L/g)	0.0102	0.0333	0.7820
	R^2^	0.9507	0.9449	0.8977
Harkins–Jura	A	0.2575	0.2683	28.490
	B	3.8835	3.7272	0.0351
	R^2^	0.8454	0.8430	0.9992
Jovanovich	q_max_ (mg/g)	0.5730	0.5871	4.3466
	K_J_	0.0317	0.0316	0.0176
	R^2^	0.8395	0.8366	0.9744
Dubinin–Radushkevich	*q_D_* mg/g	2.7977	0.3270	8.5165
	*B* mol^2^/kJ^2^	4 × 10^−6^	4 × 10^−6^	1 × 10^−5^
	E (kJ/mol)	353.55	353.55	223.61
	R^2^	0.8768	0.8895	0.6956

## Data Availability

The original contributions presented in the study are included in the article/Appendix A, further inquiries can be directed to the corresponding author.

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
