# Peer review of "Nitrogen Removal from Landfill Leachate Using Biochar Derived from Wheat Straw"

_materials, 2024, doi:10.3390/ma17040928_

Round 1
Reviewer 1 Report
Comments and Suggestions for Authors
This is an interesting paper. I have only few minor suggestions:
1. Please indicate what the two curves in Fig. 5 correspond to. The legend to y-axis should be in English.
2. Could the two plots in Fig. 6 be placed in a single plot? Comparison of results would be much better then.
3. One decimal point should be enough in all numbers in Fig. 7.
4. It seems that similar BC (from wheat straw) has been studied for adsorption capacity in the removal of nitrogen. In that case a comparison with other results would be appropriate.
5. A possible typo in the Abstract - in 93.11 a '%' is missing?
Comments on the Quality of English LanguageN/A
Reviewer 2 Report
Comments and Suggestions for Authors
The manuscript deals with an interesting research topic, having a sound methodology, but weakly explanatory interpretation of findings. Besides, there is need for specific non-citing parts to be better validated and verified, thus, there is room for further improvements prior the manuscript to be accepted for publication at the Materials journal. To this end the following review comments can be considered.
1. Data of Table 1 can be accompanied by relevant citations in which measurements-analyses took place.
2. The analytical processes and the experimental session of section 2, and all its subsections 2.1-2.4, have to be accompanied by relevant citations, respectively. Supportive and relevant to large-scale applications of adsorbents to pollutants removal, have to be added in order to verify and to validate the employed methodology and derived outcomes.
DOI: 10.5004/dwt.2019.24909 , https://www.deswater.com/DWT_articles/vol_170_papers/170_2019_111.pdf
DOI: 10.1504/IJETM.2006.010482
DOI: 10.1504/IJETM.2006.010482
DOI: 10.1080/15422110600822733
Besides, these citations can also reinforce the citations of the Introduction, offering a more concrete and pluralistic theoretical overview of relevant applied methodologies and analyses globally.
3. The explanatory text of Table 2 and Figures 4a, 4b, has to go after their graphs-Table2, no before them. Therefore, 2-3 introductory text before inserting these graphs-Table2 into the main text of subsection 3.1. This Figure-first, explanatory text-second, order has to follow all Tables and Figures throughout the study, which are currently placed in a messy manner. No dense arrangement of Figures and Tables one after the other without explanatory text among them is permissible.
4. In subsection 3.1 “FTIR analysis provides valuable information regarding the chemical composition and functional groups present in materials. These will offer insights into the potential interactions and adsorption mechanisms between the AC and BC materials and the adsorbate, nitrogen”, but it is not clearly conveyed from the existing graphs. Therefore, authors are recommended to accompany this statement with relevant graphical representation and narrative explanation, respectively.
5. Next to subsection 3.4 authors are recommended to add a separate Discussion section, in which all findings to be conveyed in an integrated-holistic manner. For this, the technological constraints, limitations, environmental concerns and technological challenges, to be collectively presented in a descriptive manner. For this, 3-4 paragraphs are adequate. This is a critical comments since all section 3 is related to the “Results” part of the study, whereas the “Discussion” part has to be be more strikingly developed.
6. In data of Table 3 if authors could accompany the isotherms and their kinetics with a pictorial-graphical-schematic representation of the developed bonds’ reforming in-between the adsorbate-adsorbent molecules’ interface, should be perfect. If not, (without need authors to be extensively detailed) just a bond-shaped narrative analysis of the molecules’ packing arrangement in the adsorbents’ sites/interface, as well as the repulsion-attractive-Van der Walls interactions developed, can all add value to the whole Discussion section. 3-4 sentences are adequate.
Reviewer 3 Report
Comments and Suggestions for Authors
The manuscript by Igwegbe et al., attempts to the use of biochar for the treatment of landfill streams. The topic slightly fits within the Materials fields of interest. However, I could not overcome the sense that there are several issues that should addressed to improve the quality of the manuscript prior to publication. In some cases, the manuscript is hastily finished/submitted, please crosscheck.
Title. There is also a very large reference to sustainability and its related terms (title, abstract, keywords and main text). However, sustainability is the triple bottom of line that equates economic, social and environmental. The paper does not refers to economic, environmental and societal pillars to support the term. Please revise accordingly.
Abstract:
What is the big conclusion, the take-home message?
The statement that “C offers a promising, environmentally friendly, and cost-effective approach for LLCH disposal, despite LLCH's complex matrix” is dubious since both life cycle and techneconomic assessments are missing, respectively to support it at hand.
Keywords: Please avoid using words already given in the Title.
Introduction:
What is novel/new in this study, compared with previous studies of the research group or other similar studies? In the introduction section, the authors need to point out how this study is different from the other limited literature (in brief). This difference will provide the motive for the study.
Materials and Methods: This section is the heart of the research. However, as it stands now crucial information concerning the employed methods are not presented and lack of meaningful analysis/interpretation of the results is noticeable. Origin of wheat straw, AC, their characteristics, equipment used for biochar production, analyses methods were poorly described, leading to at best questionable usefulness of the presented data.
Table 1. SI units only e.g. mg/L instead of mg/l
Table 1. Measurement uncertainty seems very high. Please crosscheck.
No information is provided on the QA/QC of the chemical analyses obtained.
Table 1 is given twice i.e.at p.4 and p.8
Results and discussion: The presentation of the results and conclusions is weak, confusing, unstructured and partially repetitive. Since the description and labelling of tests/experiments is not given throughout the paper, I was not able to follow the overall approach and to understand the conclusions. For example, I am not even sure how many tests were performed or why the theoretical background i.e. Table 3 is given at the end of the manuscript.
Conclusions: The conclusions section is weak given the novelty of the data. It is just a repetition of the abstract. The vast majority of data/results scattered in the text are not reiterated in the conclusions. For this paper to be of value to a broader audience, an expended conclusions section is required. These expanded conclusions need to focus upon the significance of the work.
Comments on the Quality of English Languagesee report
Round 2
Reviewer 2 Report
Comments and Suggestions for Authors
At this revised manuscript authors proceeded in a substantial improvement of their initial study, having the review comments addressed in a systematic and meticulous manner. All research parts, analysis and findings, have been fully developed, disclosing insightful findings in the fields of environmental remediation and the decisive role of waste management in mitigating environmental pollution. In this respect the revised manuscript can be accepted for publication at the Materials journal as is.
Reviewer 3 Report
Comments and Suggestions for Authors
- The revised version reads well. Authors have addressed all the comments raised in the last review. However, still there are editorial irregularities which I believe can be corrected at the typesetting and proof stage. Thus, this manuscript can now be accepted for publication.
Comments on the Quality of English LanguageThe revised version reads well. Authors have addressed all the comments raised in the last review. However, still there are editorial irregularities which I believe can be corrected at the typesetting and proof stage. Thus, this manuscript can now be accepted for publication.